# OpenReview forum: "Multimodal Function Vectors for Visual Relations"
_ICML.cc/2026/Conference — ICML 2026 regular_

### Official Review · Reviewer_BCqo · 2026-03-12

**Soundness:** 4
**Presentation:** 3
**Significance:** 3
**Originality:** 3
**Overall Recommendation:** 4
**Confidence:** 4

**Summary:**

This paper finds only a small subset of attention heads in large multimodal models are responsible for transmitting representations of spatial relations in multimodal in-context learning. Furthermore, the activation of these attention heads can be extracted and manipulated to alter an LMM’s performance on relational tasks. Through experiments on two LMMs, the authors demonstrated that the internal behavior of these models can be steered and optimized.

**Compliance With Llm Reviewing Policy:**

Affirmed.

**Key Questions For Authors:**

Please refer to weaknesses

**Limitations:**

yes

**Strengths And Weaknesses:**

Strengths:
The findings in this paper makes the internal mechanism of LMMs more transparent, which is helpful for further diagnosing their failures, improving their performance and making them more reliable. Through early experiments, the authors show that the multimodal function vectors can be fine-tuned to steer the behaviour of the model, which is stronger than direct in-context learning, and is more parameter efficient than directly fine tuning the model.

Weaknesses:
1. The experiment scope is limited. Object relationships in synthetic dataset and GQA are both mostly solved by current sota LMMs. In addition, there are some fields that are still very challenging for LMMs, such as mental rotation. Including more challenging tasks to the experiment would greatly strengthen this paper.
2. More detailedly and grounded discussion on the broader impact is expected. For example, how can the steering method (finetuning FV) in this paper be applied to general LMM training? Is the insight from this paper potentially helpful to more LMM applications other than multimodal in context learning and relational reasoning?

---

> ### Author Rebuttal · Authors · 2026-03-31
>
> **1.** We respectfully disagree with the claim that these tasks are “mostly solved.” Our experiments show that even a state-of-the-art model such as Qwen3-VL achieves only ~31.6% accuracy on GQA images in a 4-shot setting (Table 1, page 13), which remains far below human-level performance. With fine-tuned function vectors (FFV), we achieve substantially higher accuracy in a zero-shot setting (Qwen3-VL + FFV: ~41.5%; see Figure 6b).
>
> More importantly, our work is not intended as a benchmark for relation detection, but as a mechanistic study of relational inference. We demonstrate, for the first time, that multimodal spatial relations are encoded in localized internal structures (function vectors) that can be extracted, fine-tuned, and even linearly composed to solve novel analogy tasks involving unseen relations (Section 4.4).
>
> ---
>
> **2.** We thank the reviewer for the suggestion to further clarify the broader impact. We believe our findings on multimodal function vectors (FVs) provide several insights with implications for general LMM/VLM training and applications:
>
> - **Modular fine-tuning.** Our results show that specific cognitive capabilities (e.g., relational reasoning) can be optimized within a localized, low-dimensional subspace (the function vector), while keeping the backbone model frozen. This suggests a modular alternative to traditional fine-tuning: instead of updating billions of parameters, one could learn a library of specialized *skill vectors* (e.g., spatial reasoning, counting, attribute comparison) that can be injected into a base model on demand, significantly reducing compute and storage costs.
>
> - **Improved efficiency.** Standard LMMs/VLMs often require multiple in-context demonstrations to activate a task, which consumes context window space and increases inference latency. In contrast, a fine-tuned FV can elicit the same behavior in a zero-shot manner. This suggests that complex behaviors could be “baked in” by directly optimizing internal circuits, rather than relying solely on prompting or large-scale instruction tuning.
>
> - **Controllability and safety.** The observation that task knowledge is localized to specific attention heads provides a pathway for targeted model control. For example, analogous to extracting a “spatial relation” vector, one could identify vectors associated with toxicity or bias, and use them for both detection and active steering (e.g., injecting an “anti-bias” vector at inference time) without retraining the full model.
>
> - **Compositional generalization.** The ability to linearly combine function vectors to solve novel composite relations (e.g., combining “above” and “left” to form “above-left”) has important implications for robotics and embodied AI. An agent could learn a set of primitive relational vectors and compose them to interpret novel instructions, without requiring explicit training on every possible combination.

---

> > ### Author Rebuttal · Reviewer_BCqo · 2026-04-04
> >
> > The poor performance of Qwen3-VL on GQA in this paper is concerning.
> > From Qwen-VL's tech report (very first version model), the model's performance is already 59.3% on GQA official dataset. This strongly suggests either the model evaluation setting or the questions are ill-posed in this paper. We can barely find any recent released models are tested on GQA (e.g. Qwen2-VL already excludes GQA), which is the reason I said it's a mostly solved problem and expansion to more challenging domains are expected.

---

> > > ### Author Response · Authors · 2026-04-05
> > >
> > > We would like to clarify that our evaluation on GQA does not follow the standard GQA VQA setting, therefore, the reported results are **not directly comparable** to those in the Qwen-VL technical report.
> > >
> > > In the standard GQA benchmark, the task is conventional VQA: the input question explicitly specifies the target relation (e.g., “What is on top of the table?”), and the model only needs to identify the corresponding object. In this case, the relation is given, and the task reduces to grounded recognition and reasoning.
> > >
> > > In contrast, our task is fundamentally different. We construct a **relational inference task**, where the model is provided with multiple in-context examples that implicitly share a relation (see Figure 3 in our manuscript), and must:
> > >
> > > **1. infer the underlying relation from the context**, and
> > > **2. apply this inferred relation to a new query object.**
> > >
> > > As described in Section 3.2, the model must infer the correct object label from the relational context and the query object. Crucially, the relation is never explicitly provided in the prompt, making this a substantially more challenging problem than standard GQA.
> > >
> > > Therefore, our setting evaluates **relation representation and compositional generalization**, rather than direct VQA performance. The relatively low accuracy reflects the inherent difficulty of learning latent relational structure via in-context learning, rather than an issue with the evaluation setup or question quality.
> > >
> > > We also note that recent models (e.g., Qwen2-VL) no longer report GQA results, likely because standard GQA VQA has become saturated. In contrast, our work leverages GQA images as a source of realistic relational scenes while introducing a new task that probes a fundamentally different and more challenging capability.

---

### Official Review · Reviewer_f4zC · 2026-03-13

**Soundness:** 2
**Presentation:** 2
**Significance:** 2
**Originality:** 2
**Overall Recommendation:** 3
**Confidence:** 4

**Summary:**

This paper explores spatial relation understanding in LMMs using function vectors. By applying causal mediation analysis on synthetic and real datasets, the authors identify critical attention heads and extract multimodal function vectors. They demonstrate that fine-tuning these vectors improves zero-shot performance. Moreover, the study shows that these vectors can be composed to solve one-shot analogy tasks with unseen relations, demonstrating robust generalization and outperforming standard in-context learning baselines.

**Compliance With Llm Reviewing Policy:**

Affirmed.

**Final Justification:**

My primary concern remains the limited scope and evaluation of this work, a point that has been raised by other reviewers. Given these persistent limitations, I have decided to raise my rating to 3.

**Key Questions For Authors:**

Please refer to the weaknesses part.

**Limitations:**

yes

**Strengths And Weaknesses:**

Strengths:

1.	The work employs a well-founded methodology to investigate spatial relation understanding in LMMs through function vectors, building upon concepts broadly explored in previous work on language models.
2.	This work demonstrates that sparse attention heads encode spatial relational knowledge within localized internal structures, which can be systematically extracted and optimized for enhancing relational reasoning.
3.	The paper is well-organized and clearly written.

Weaknesses:

1.	The core idea relies on function vectors and activation editing, which have been extensively explored in prior works. While the application to spatial relation is valuable, the fundamental technique represents more of an extension than a novel theoretical contribution.
2.	The evaluation is confined to elementary spatial prepositions (e.g., above, below, left, right). How can the authors demonstrate that their approach generalizes to the complex, compositional, and often ambiguous relations encountered in real-world multimodal reasoning?
3.	The experiments are primarily conducted on controlled benchmarks. More discussion or experiment results regarding the method's performance in OOD noisy, real-world environments would strengthen the claims about its practical utility.

---

> ### Author Rebuttal · Authors · 2026-03-31
>
> **1.** We thank the reviewer for this insightful comment and agree that function vectors and activation editing have been explored in prior work, particularly in language-only settings. Our contribution is not to introduce a fundamentally new mechanism, but to extend and rigorously study this paradigm in the significantly more complex setting of multimodal spatial relational reasoning.
>
> Importantly, this extension is non-trivial. Large multimodal models integrate high-dimensional visual features with language representations, and it is not obvious that compact, reusable function vectors can still be extracted or retain causal control in such a fused space. Our results show that they can. Moreover, we find that spatial relational knowledge, despite not being directly observable from pixels, can be localized to a sparse subset of attention heads and manipulated through these vectors.
>
> We also demonstrate new capabilities not shown in prior work: function vectors can be fine-tuned while keeping the backbone frozen, substantially improving performance over in-context learning, and relation-specific vectors can be composed to solve analogy tasks involving unseen relations, indicating compositional generalization.
>
> In addition, prior work on function vectors in LLMs constructed uninformative contexts using randomly shuffled word pairs for causal intervention analyses. We propose an alternative approach that preserves consistency between images and text while still creating uninformative contexts, which constitutes a new contribution.
>
> We position our work as a principled extension of function vectors to multimodal models that reveals how spatial relational knowledge is internally represented and enables new forms of control and generalization. We will revise the paper to better clarify this distinction between building on prior techniques and providing new empirical and conceptual insights.
>
> ---
>
> **2.** We extend our evaluation beyond spatial relations to more complex agentic relations (e.g., *carrying*, *holding*, *riding*, *wearing*), which better reflect real-world multimodal reasoning. We construct a dataset of 1,652 images and evaluate using the same setup as in the main paper.
>
> As shown in Table 1 of our response to Reviewer m5GB, function vectors consistently outperform both zero-shot and in-context learning baselines, with fine-tuned function vectors achieving the largest gains (54.8% vs. 34.4% for 4-shot ICL).
>
> We refer the reviewer to our response to Reviewer m5GB (Point 1) for additional details.
>
> ---
>
> **3.** We agree that evaluating robustness in more realistic and noisy settings is important for assessing practical utility. While our synthetic dataset provides a controlled environment for precise causal analysis, we also include experiments on the GQA dataset (https://huggingface.co/datasets/lmms-lab/GQA), which consists of real-world images with diverse scenes and more complex relational structures. In the current paper, we present one example from GQA in Figure 2b and the results on GQA in Figure 6b. In the revision, we will include additional qualitative and quantitative examples using GQA images. Notably, our method shows consistent improvements over both zero-shot and in-context learning baselines on this dataset, suggesting that the extracted function vectors capture relational knowledge that transfers beyond controlled settings.
>
> In addition, we evaluate generalization in several out-of-distribution scenarios. First, we test on held-out objects not seen during training and observe that fine-tuned function vectors maintain strong performance, indicating that the learned representations are not tied to specific object identities (see Section 4.3). Second, we demonstrate cross-dataset transfer, where function vectors extracted from synthetic data achieve comparable performance when applied directly to real-image GQA tasks, further supporting robustness across domains (see Appendix G and Figure 14).
>
> Overall, these results demonstrate meaningful robustness beyond controlled settings: improvements on real-image data, generalization to unseen objects, and successful cross-dataset transfer all indicate that the learned function vectors capture relational structure in a way that is not tied to specific datasets.

---

> > ### Author Rebuttal · Reviewer_f4zC · 2026-04-02
> >
> > Thank you for the comprehensive rebuttal. However, my primary concern remains the limited scope and evaluation of this work, a point that has been raised by other reviewers. Given these persistent limitations, I have decided to raise my rating to 3.

---

> > > ### Author Response · Authors · 2026-04-05
> > >
> > > Thank you for your thoughtful follow-up and for taking the time to carefully consider our rebuttal. We appreciate your recognition of the revisions and your decision to raise the score.
> > >
> > > We understand your concern regarding the current scope of the evaluation. As also acknowledged in the paper, our study focuses on a controlled set of spatial relations to enable precise causal analysis, which naturally limits coverage of more complex real-world scenarios. That said, we agree that broader and more diverse evaluations are important, and we have already taken steps in this direction (e.g., extending to more complex relations and real-image datasets such as GQA). In the revision, we will further expand these results and clarify the scope and limitations more explicitly.
> > >
> > > More broadly, our goal is to establish a principled and interpretable framework for studying relational representations in LMMs, with a goal to show that function vectors can be extracted, manipulated, and composed in a multimodal setting . We view this as a foundation that can be extended to richer and more realistic settings in future work, and we will better emphasize this positioning in the paper.
> > >
> > > Thank you again for your constructive feedback. It has been very helpful in improving both the clarity and positioning of our work.

---

### Official Review · Reviewer_m5GB · 2026-03-16

**Soundness:** 3
**Presentation:** 4
**Significance:** 4
**Originality:** 3
**Overall Recommendation:** 5
**Confidence:** 4

**Summary:**

This manuscript investigates the internal mechanisms of Large Multimodal Models in representing spatial relations. Drawing on the function vector framework from language modeling, the authors employ causal mediation analysis to identify a sparse subset of attention heads responsible for transmitting relational knowledge. The study demonstrates that these multimodal function vectors can be extracted and fine-tuned to elicit correct spatial predictions in a zero-shot setting, significantly outperforming standard in-context learning baselines. Furthermore, the paper validates the linear representation hypothesis in multimodal space, showing that base relation vectors can be linearly combined to solve analogy problems involving novel, untrained relations. Through extensive experiments on synthetic and real-world datasets like GQA, the work shows that extracted relational representations are modular, transferable across domains, and causally influential model behavior.

**Compliance With Llm Reviewing Policy:**

Affirmed.

**Final Justification:**

Based on the overall assessment, I maintain my score.

**Key Questions For Authors:**

1.If the authors demonstrate that the framework is applicable to non-spatial or abstract relations while maintaining head sparsity, it would significantly increase the paper's Significance.

2.Evidence that composition weights can be inferred without a source analogy would greatly enhance the Originality and practical utility of the method.

3.A clearer explanation of why relational geometries differ between backbones would strengthen the Soundness of the claims regarding localized representations.

**Limitations:**

yes

**Strengths And Weaknesses:**

The submission offers an original extension of mechanistic interpretability tools to the multimodal domain, addressing a significant gap in our understanding of how Large Multimodal Models process task-specific knowledge. A major strength is the rigorous use of causal mediation analysis to localize specific circuits, providing solid evidence for the existence of specialized "relational heads." The paper is well-presented, with clear visualizations of the vector extraction and composition process. However, the work has several weaknesses that limit its overall impact. The evaluation is restricted to a narrow set of basic spatial relations, leaving the framework’s applicability to more abstract or complex physical and social relations unexplored. There is also a substantial performance gap between the fine-tuned function vectors and human-level accuracy, suggesting that the extracted vectors may only capture a partial representation of the underlying reasoning. Additionally, the reliance on a source analogy to infer composition weights for composite function vectors limits the method's autonomy. The negative representational similarity analysis correlations between different backbones raise questions regarding the universality of the identified relational geometries across different model architectures.

---

> ### Author Rebuttal · Authors · 2026-03-31
>
> **1.** We agree that demonstrating generalization beyond spatial relations is important for establishing the broader applicability of our approach to real-world multimodal reasoning. While our initial experiments focus on controlled spatial relations, we extend our evaluation to more complex agentic relations, including *carrying*, *holding*, *riding*, *sitting on*, *standing in*, *standing on*, and *wearing*. These relations better reflect real-world scenarios, as they involve compositional structure and greater semantic ambiguity.
>
> To support this analysis, we construct a new dataset of 1,652 images covering these agentic relations and evaluate OpenFlamingo using the same pipeline as in the main paper. The average performance across all agentic relations is summarized below:
>
> **Table 1. Top-1 prediction accuracy on agentic relation tasks.**
>
> | 0-Shot | 4-Shot | Initial FV | Fine-tuned FV |
> |--------|--------|------------|----------------|
> | 0.177  | 0.344  | 0.226      | 0.548          |
>
> We observe that function vectors consistently improve performance over both zero-shot and in-context learning baselines. In particular, fine-tuned function vectors yield the largest gains. On average, OpenFlamingo achieves 34.4% accuracy with 4-shot in-context learning, while fine-tuned function vectors reach 54.8% accuracy in a zero-shot setting on agentic relations. These results suggest that function vectors are not limited to spatial reasoning, but instead capture more general relational patterns that extend to semantically richer and more complex interactions.
>
> ---
>
> **2.** We thank the reviewer for raising this important point. We intentionally formulate composite function vectors within a one-shot analogy setting, rather than a fully autonomous zero-shot composition setting. Inferring composition weights for relations without any contextual signal is fundamentally underdetermined: without at least one example of how base relations combine, there is no principled way to assign weights to the constituent vectors.
>
> The source analogy in the one-shot setting provides this minimal signal, enabling the model to infer the relative contributions of each relation from a single demonstration, rather than relying on hand-crafted rules. In this sense, the source analogy acts as a lightweight mechanism for reading out compositional structure, rather than a limitation of the approach.
>
> As noted in the paper, alternative formulations are possible. For example, one could impose fixed or learned composition rules (e.g., equal-weight combinations), but such approaches bypass the core of analogy reasoning—namely, inferring and transferring relational structure from a familiar domain to a new one. In contrast, our formulation shows that a single demonstration is sufficient to infer composition weights and generalize effectively, which we view as a meaningful step toward more autonomous relational reasoning.
>
> ---
>
> **3.** We believe this discrepancy is likely driven by architectural and training differences between the models, particularly in how multimodal information is fused and aligned. Models with stronger vision-language alignment and more advanced fusion mechanisms (e.g., Qwen3-VL) may develop more distinct subspaces for different relations, whereas earlier architectures (e.g., OpenFlamingo) may rely on more shared representations.
>
> Importantly, despite these geometric differences, both models exhibit localized, causally influential subsets of attention heads for relational reasoning. This suggests that while the global structure of the representation space may differ, the presence of sparse, functionally meaningful subnetworks is a more general property of vision-language models for capturing visuospatial relations.

---

> > ### Author Rebuttal · Reviewer_m5GB · 2026-04-03
> >
> > Thank you for the authors' replies. Their responses addressed many of my concerns. However, is there quantitative experimental evidence to support the claim that strongly aligned models form more independent subspaces?

---

> > > ### Author Response · Authors · 2026-04-06
> > >
> > > Thank you for this helpful follow-up question. We agree that our original explanation was primarily qualitative, and we appreciate the opportunity to provide more concrete evidence and clarify the scope of our claim.
> > >
> > > To investigate whether different backbones organize relational representations differently, we analyzed the **effective rank** of the function vector sets for each model. Intuitively, effective rank measures how many independent directions are present in a set of vectors:
> > > - A **higher effective rank** indicates that the vectors span a more independent subspace.
> > > - A **lower effective rank** suggests that the vectors are more tightly clustered or overlapping.
> > >
> > > Concretely, we compute the effective rank of the matrix formed by stacking the relation-specific function vectors. The results are:
> > >
> > > | Model        | Initial FV | Fine-tuned FV |
> > > |--------------|-----------:|--------------:|
> > > | OpenFlamingo | 2.436      | 2.777         |
> > > | Qwen3-VL     | 3.798      | 3.829         |
> > >
> > > These results suggest two consistent patterns:
> > >
> > > - **Qwen3-VL exhibits a higher effective rank than OpenFlamingo**, indicating that its relation vectors occupy a more expanded and differentiated subspace.
> > > - **Fine-tuning increases the effective rank** in both models, suggesting that optimization encourages greater separation between relation representations.
> > >
> > > This quantitative pattern is consistent with our qualitative observations from both performance results (Figure 6) and representational similarity analysis (Appendix H). In particular, OpenFlamingo function vectors are highly clustered (pairwise cosine similarities > 0.9) and achieve lower performance, whereas Qwen3-VL vectors are more separated and yield stronger performance.
> > >
> > > Importantly, we would like to clarify the scope of our claim. We do **not** intend to argue for a causal relationship between model strength (or alignment) and subspace independence. Specifically:
> > >
> > > - The negative cross-model RSA results indicate that relational geometries are **not shared across architectures**, but do not establish any ordering or causality.
> > > - The effective rank analysis should therefore be interpreted as **descriptive evidence within the models we study**, rather than a universal or causal property.
> > >
> > > We will revise the paper to (i) include this effective rank analysis, (ii) clarify its interpretation, and (iii) explicitly frame the result as a correlational observation rather than a causal explanation.
> > >
> > > We thank the reviewer for prompting this clarification and helping strengthen the rigor of our claims.

---

### Decision · Program_Chairs · 2026-04-30

**Decision:**

Accept (regular)

**Comment:**

This paper explores spatial relation understanding in LMMs using function vectors. After rebuttal, it received scores of 534. While reviewers noted some limitations in the scope of evaluation and generalization, they agreed that the paper makes a meaningful and technically sound contribution by extending mechanistic interpretability to multimodal models, with rigorous causal analysis and insightful identification of relational circuits. Therefore, the paper can be accepted to the conference.